# Evaluation of Haemostasis in Dogs Affected by Resectable Malignancy

**DOI:** 10.3390/ani13010164

**Published:** 2022-12-31

**Authors:** Barbara Bruno, Cristiana Maurella, Paola Gianella, Giulio Mengozzi, Erica Ferraris, Antonio Borrelli

**Affiliations:** 1Department of Veterinary Science, University of Turin, Largo P. Braccini 2, 10095 Grugliasco, Italy; 2Section of Turin, Istituto Zooprofilattico Sperimentale del Piemonte, Liguria e Valle D’Aosta, Via Bologna 148, 10154 Turin, Italy; 3Department of Public Health and Pediatric Sciences, C.so Bramante 88/90, 10100 Turin, Italy

**Keywords:** coagulation, thromboelastometry, thrombin generation, canine, neoplasia

## Abstract

**Simple Summary:**

In human cancer patients, recent studies have focused on identifying hypercoagulability, which was observed before surgery and may occur after surgery and persist postoperatively. Even in veterinary medicine, dogs with neoplasms may be hypercoagulable and considered to be at risk for thrombosis. The aim of this study was to observe the characteristics of haemostasis in dogs undergoing surgery for the removal of malignancies using tools that allow a more comprehensive assessment of coagulation (rotational thromboelastometry and thrombin generation). Haemostasis was evaluated immediately before surgery (T0), 24 h after surgery (T1), and two weeks after surgery (T2). Statistically significant differences were found between some rotational thromboelastometry parameters at T0 vs. T1 and at T1 vs. T2, indicating a trend towards hypercoagulability 24 h after surgery and a reversal of the trend 2 weeks after surgery. The results of this study observed a transient influence of surgery on haemostasis and may help us to understand how long anticoagulant treatment should be administered in these dogs.

**Abstract:**

Dogs with neoplasm are at risk of thrombosis, yet there is no information on the haemostatic alterations that may result from the surgeries performed to remove neoplastic masses. The aim of this study was to observe the characteristics of haemostasis in dogs undergoing surgery to remove a malignancy by means of rotational thromboelastometry and thrombin generation. Haemostasis was assessed immediately before surgery (T0), 24 h after surgery (T1), and two weeks after surgery (T2). Statistically significant differences were found between the thromboelastometric parameters at T0 vs. T1, with increases observed in MCF, the α angle, and G parameters in the ex-TEM and in-TEM profiles. In the thromboelastometric analysis performed after surgery differences were identified between T1 and T2, with a decrease observed in parameters such as CFT, MCF, the α angle, and G. Between T0 and T2, only a significant decrease in CT was detected in the fib-TEM profile. No differences were found in the comparison among the results obtained via thrombin generation. In dogs with resectable malignancies, the surgeries performed to remove cancer affected coagulation, causing a tendency towards hypercoagulability. The interference with coagulation was transient, and 2 weeks after surgery, the thromboelatometric results returned to those obtained before surgery (T0).

## 1. Introduction

Hypercoagulability is an alteration in haemostasis resulting from an imbalance between procoagulant and anticoagulant factors and that occurs in several pathological conditions. The presence of thrombophilia is a risk for thrombi formation, and the identification of this haemostatic dysregulation by viscolelastic tools helps to guide prophylactic therapy [1,2].

In veterinary medicine, the CURATIVE guidelines have identified dogs with neoplasms as patients who are at risk for thrombosis, particularly in cases of sarcomas, carcinomas, and lymphomas, with the administration of antithrombotic therapy being recommended if hypercoagulability is detected or if it is in the presence of a risk factor associated with thrombosis [1].

In human cancer patients, the presence of thrombophilia has long been recognized, especially in some certain malignancies, and deep vein thrombosis is one of the most common complications [3,4]. Moreover, the risk of pathological thrombosis often appears to be exacerbated during cancer treatment [5].

There are some hypotheses to explain the activation of the coagulation system in malignancies, many of which involve the tumour-induced activation of platelets, the increased expression of tumour-related intravascular tissue factor (TF), and the upregulation of endogenous inhibitors of fibrinolysis [6]. Surgery is also a stressful event, involving inflammation, ischemia–reperfusion injury, and cytokine release [7].

The use of viscoelastic tools such as the thomboelastometer or thromboelastograph allows for a more complete assessment of haemostasis and for the identification of hypercoagulability, a condition that is difficult to detect using other methods [8]. In human cancer patients, some recent studies have focused on the identification of hypercoagulability, and it has been observed that in some cases, this haemostatic alteration was present already before surgery and occurs after surgery, persisting for at least 1 month postoperatively before returning to baseline levels at 6–12 months [3,4,9]. Another observation reported in the study of Muller et al. (2019) was that hypercoagulability developed in patients undergoing more invasive types of surgery (e.g., open surgery for esophagectomy) [10].

Additionally, in veterinary medicine, some prospective studies have detected thrombophilia in dogs affected by malignancies, showing that this condition persists in dogs with lymphoma during treatment [11,12,13,14]. In canine patients, however, there is no information on the haemostatic changes that might occur after surgery performed to remove a neoplastic mass.

High thrombin levels in cancer patients not only imply a state of hypercoagulability and a consequent increased risk of cancer-associated thrombosis, but also promote tumour growth and metastasis [5]. The thrombin generation assay (TG) is a global dynamic test that estimates the overall potential of the haemostatic system by simultaneously and continuously measuring thrombin generation [15]. Elevated thrombin generation, which is detected in human beings affected by malignancies, has been found to be associated with an increased risk of venous thromboembolism [16,17]. In veterinary medicine TG has been used to assess coagulation in dogs with hyperadrenocorticism or acute gastroenteropathy, with significant differences being observed between healthy and sick dogs in both studies, with an increase in thrombin generation being observed in the latter group [18,19].

The aim of this study was to observe the characteristics of haemostasis in dogs undergoing surgery for the removal of malignancies by means of rotational thromboelastometry and thrombin generation. Our first hypothesis was that in dogs affected by neoplasia, surgery acts as additional stimulating factor on coagulation, causing hypercoagulability and increasing the risk of thrombosis.

If alterations in haemostasis were detected after surgery, the second hypothesis was that these alterations would persist two weeks after surgery.

## 2. Materials and Methods

### 2.1. Animals

The study protocol was approved by the Institutional Ethics and Animal Welfare Committee (protocol number 1128). It was a prospective investigation that involved client-owned dogs, and all owners provided written informed consent. All cases enrolled were adult dogs admitted to the Veterinary Teaching Hospital, and all dogs had a neoplasm diagnosis (sarcomas, carcinomas, and lymphomas) as well as planned surgery for removal (thoracotomies, laparotomies, complex skin plastics, mandibulectomies, and maxillectomies). Haematocrit < 25%, body weight less than 15 kg, and a remote history of spontaneous bleeding and/or the administration of blood products during surgery were considered exclusion criteria.

At presentation, clinical data were collected, including recent and remote history, and a complete physical examination was carried out. Whole blood samples were collected to perform the following laboratory analyses: complete blood count (ADVIA 120 Hematology, Siemens Healthcare Diagnostics, Erlangen, Germany) with blood smear evaluation (CBC), serum chemistry profile (including measurement of sCr, urea, albumin, glucose, alkaline phosphatase, aspartate aminotransferase, alanine aminotransferase, and γ-glutamyl transpeptidase) (ILAB 300 plus, Clinical Chemistry System, Instrumentation Laboratories, Midland, ON, Canada), venous blood gas analysis (including lactate and electrolyte concentrations) (ABL 800 Flex; A. DE MORI S.p.A.); packed cell volume (PCV) and total proteins (TP); and urinalysis, including urine specific gravity (USG) (Reichert VET 360, Reichert technologies analytical instrument, Depew, NY, the USA), dipstick examination (Multistix 10 SG Reagent Strips, Siemens Healthcare Diagnostics, Erlangen, Germany), microscopic evaluation of the urine sediment, and urine chemistry, including urine protein to urine creatinine ratio (UPC). Serological tests for Leishmania infantum, Ehrlichia canis, Borrelia burgdorferi, Anaplasma phagocytophilum, and Dirofilaria immitis were performed (Snap Leishmania and 4 Dx, IDEXX Laboratories, Westbrook, Maine, United States).

A haemostasis assessment was performed using the standard coagulation profile [prothrombin time (PT), activated partial thromboplastin time (aPTT) and fibrinogen] (Coagulometer StART, Diagnostica Stago), rotational thromboelastometry (ROTEM, Tem International GmbH, Munich, Germany) and thrombin generation (Thrombinoscope, Stago, Asnières-sur-Seine, France).

### 2.2. Assessment of Haemostasis

Haemostasis was assessed immediately before surgery (T0), 24 h after surgery (T1), and two weeks after surgery (T2).

Samples were obtained by jugular venipuncture (using a 20-gauge needle), and when venipuncture needed to be repeated, the needle was repositioned, and/or blood flow was interrupted; the blood was discarded; and samples were taken from the contralateral jugular vein. The collected whole blood was placed into two tubes containing 3.2% trisodium citrate (1-part citrate: 9-parts blood).

Thromboelastometric analyses were performed according to PROVETS guidelines, and analyses were run at least for 30 min [20,21]. Citrated blood samples were stored at room temperature and were analyzed within 30 min from blood collection. For each sample, in-TEM, ex-TEM, and fib-TEM profiles were performed to evaluate the intrinsic pathway (activation by ellagic acid), the extrinsic pathway (tissue factor activation), and functional fibrinogen (platelets inactivated with cytochalasin D), respectively. For each profile, the following parameters were assessed: clotting time (CT, s), clot formation time (CFT, s), maximum clot firmness (MCF, mm), α angle (α, °), and G, the value of which was calculated from measurements taken to determine the total clot strength [G = 5000 × MCF/(100-MCF)]. The profiles are graphically represented as curves (Figure 1). The first fibrin clot that formed from the onset of test activation until the clot reached 2 mm in width is labeled as CT; this clot is affected by the concentration of plasmatic factors and the coagulation inhibitors (e.g., antithrombin or anticoagulant drugs) [22,23]. The velocity of clot formation was represented by CFT and is influenced by platelet count and function as well as by fibrinogen activity. The maximum firmness achieved by the clot is the MCF; the MCF is determined by the platelet count as well as by function and fibrin formation in the presence of factor XIII [20,21]. The slope of the tangent on the elasticity curve is the α angle, which describes the kinetics of clot formation, and it is affected by platelet count, function, and fibrinogen [22,23]. Hypercoagulable ROTEM tracing is characterised by a decrease in CT or CFT and an increase in MCF, G, or the α angle, whereas hypocoagulable tracing is characterised by an increase in CT or CFT and a decrease in MCF, G, or the α angle. Hypercoagulability was defined as >1 ROTEM parameters (CT, CFT, MCF, α angle) and as G being outside the upper reference range [24].

Thrombin generation (TG) (Thrombin Generation Assay and Haemostatic Profile for Elucidating Hypercoagulability in Endogenous Canine Hyperadrenocorticism) was carried out using a 96-well plate fluorimeter and citrated platelet-poor plasma that had been stored at −80 °C for less than 12 months [25]. Each test required two sets of readings, one for TG and another for thrombin calibrator measurements. Each well was pipetted with 80 µL of plasma. Next, 20 µL of thrombin calibrator reagent (Diagnostica Stago, Asniere, France) was added to the thrombin calibrator wells, and an additional 20 µL of recombinant tissue factor (TF) (PPP-reagent; Diagnostica Stago) was added into the TG wells. The filled plates were incubated in a fluorimeter for 10 min at 37 °C. Finally, 20 µL of fluorescent substrate (Fluca-Kit, Diagnostica Stago) was automatically dispensed into each well. The reaction was read continuously for 60 min using Thrombinoscope software (Diagnostica Stago), generating a TG curve (Figure 2). The following parameters were taken into account: the lagtime, which represents the initiation of coagulation (the time it takes to achieve 1/6 of the TG peak); the thrombin peak, which refers to the maximum thrombin concentration; the time to peak (TTpeak), which refers to the time to achieve the peak; and the endogen thrombin potential (ETP), which refers to the total amount of thrombin generated in the sample [26].

### 2.3. Statistical Analysis

Data were entered into an ad hoc database, analyzed with Stata 17, and tested for Normality using the Shapiro–Wilk test. To assess the differences between T0 and T1, T1 and T2, and T0 and T2, the Wilcoxon matched-pair signed-ranks test was used when the data were not normally distributed; otherwise, a Student’s t test for paired data was used. The significance level was set at *p* < 0.05.

## 3. Results

In total, 13 dogs were included in the study, but two dogs died during or immediately after surgery. The 11 dogs included were divided as follows: 7 females (5 intact and 2 spayed) and 4 males (2 intact and 2 neutered); the median age was 11 years (min 8–max 14), and the median body weight was 26 kg (min 15–max 52). The following breeds were included: mixed breed (N= 4), German shepherd (N = 2), Bernese mountain dog (N = 1), Lagotto Romagnolo (N = 1), poodle (N = 1), schnauzer (N = 1), and border collie (N = 1). The cancer types affecting the dogs were as follows: thyroid carcinoma (N = 3), leiomyosarcoma (N = 1), hepatocellular carcinoma (N = 1), liposarcoma (N = 1), fibrosarcoma (N = 1), pheochromocytoma (N = 1), adrenocortical carcinoma (N = 1), maxillary osteosarcoma (N = 1), and costal chondrosarcoma (N = 1).

The coagulation results and the results obtained for other laboratory parameters of interest are presented in Table 1, Table 2 and Figure 3. No platelet counts below the reference value were observed in any of the dogs at any time [platelets < 128 × 10^9^/L (<128 × 10^3^ cells/µL)]. Two dogs had a PPT below the lower reference range at T0 (respectively 11 and 11.7 s). The fibrinogen concentration was below the reference range in one dog at T0 (132 mg/dL) and over the range in 2 dogs at T0 (675 mg/dL) and at T1 (579 mg/dL). One dog had proteinuria at T0 (UPC 1.93). 

### 3.1. Comparison between T0 and T1

Between T0 and T1, the following statistically significant differences were found: an increase in MCF (*p* = 0.006), G (*p* = 0.02) and a decrease in CFT (*p* = 0.032) in the in-TEM profile; an increase in MCF (*p* = 0.019), the α angle (*p* = 0.035), and G (*p* = 0.006) in the ex-TEM profile. Decreases in CT (*p* = 0.033) in the fib-TEM profiles as well as in the PLT number (*p* = 0.039) and in the Hct level (*p* = 0.006) were observed from T0 and T1. No differences were found in the TG results between the two different sampling times.

### 3.2. Comparison between T1 and T2

The following statistically significant differences were observed between T1 and T2: a decrease in MCF (*p* = 0.035), the α angle (*p* = 0.019), and G (*p* = 0.035); an increase in CFT (*p* = 0.019) in the ex-TEM profile; and a decrease in MCF (*p* = 0.035) and G (*p* = 0.035) in the fib-TEM profile. No differences were found between PT and aPTT, but figrinogen levels decreased (*p* = 0.019), and the PLT number increased (*p* = 0.002) from T1 to T2. No differences were found in the data for TG between the two different sampling times.

### 3.3. Comparison between T0 and T2

Few statistically significant differences were found between T0 and T2: a decrease in CT (*p* = 0.0039) in the fib-TEM profile, a decrease in Hct level (*p* = 0.039), and an increase in the PLT number (*p* = 0.002).

### 3.4. Hypercoagulable ROTEM

At T0, four dogs were hypercoagulable. The same four dogs remained hypercoagulable at T1, whereas at T2, only two dogs were classified as hypercoagulable. In particular, one of the four dogs that was hypercoagulable at T0 had ROTEM parameters that were all above the reference ranges, denoting severe hypercoagulability (Figure 4). The TG results were also markedly increased compared to the mean values for clinically healthy dogs, indicating an increase in thrombin formation (Figure 5). This dog underwent a bilateral maxillectomy for osteosarcoma and died suddenly 36 h after surgery.

## 4. Discussion

This study monitored haemostasis in dogs affected by resectable neoplasia, performing ROTEM and TG analysis pre-surgery (T0), 24 h after surgery (T1), and 2 weeks (T2) post-surgery. The thromboelastometric analysis showed a trend towards hypercoagulability in the dogs that were included. Indeed, statistically significant differences were found between the ROTEM parameters at T0 and T1, with increases in MCF, the α angle, and G parameters being observed in the ex-TEM and in-TEM profiles. The ROTEM analysis performed after surgery identified differences between T1 and T2, with a decrease in parameters such as MCF, the α angle, and G, denoting an inverse trend compared to the differences found between T0 and T1. The repeated measurements taken 2 weeks after surgery (T2) only showed a significant decrease in CT in the fib-TEM profile compared to the data collected at T0, indicating that the hypercoagulability induced by surgery is transient and that the coagulation at T2 returns to values similar to those before surgery. No differences were found in the comparison among the results obtained with thrombin generation.

The results of the present study suggest that surgery affects coagulation and may be an additional risk factor for the development of hypercoagulability in dogs with malignancies. Indeed, the effects of surgery can contribute to the development of hypercoagulability or fuel the haemostatic alterations that are already present. The hypercoagulable phenotypes of malignancies differ from those of other pathologies, and the molecules and cells that are involved may include procoagulant factors (e.g., tissue factor), soluble mediators (e.g., adenosine diphosphate, interleukin-1, and tumour necrosis factor), and adhesion molecules (vascular cell adhesion molecules and intercellular adhesion molecules) [6].

Surgery-induced thrombophilia could be the result of severe mechanisms such as inflammation, ischemia–reperfusion injury, sympathetic nervous system activation, and increased cytokine release. The stimulus resulting from surgery affects the entire body, including tumour cells, with the awakening of dormant cells, the promotion of tumour cells proliferation, and the production of pro-metastatic factors being examples [7]. Another consequence of surgery is the haemostatic activation resulting from dysregulated coagulation and to the consequent structural and biochemical imbalances of coagulation factors, platelets, and endothelium.

The present study does not include a control group, but the authors conducted a previous study on dogs undergoing orthopaedic surgery (adult and young animals without an underlying pathology) to observe the influence of surgery on haemostasis. The blood samples used for ROTEM analysis were taken 1 h before the surgery, 24 h after the conclusion of the surgery, and 7 days after surgery. After the procedure, the ROTEM profiles showed relative hypercoagulability in the dogs, with MCF increases being observed in the in-TEM and fib-TEM profiles, denoting an influence of the functional fibrinogen concentration on significantly increased variables [27]. In the cited study, only a few ROTEM parameters changed after elective surgery, similar to in another recent study conducted on healthy dogs evaluating the effect of anaesthesia on coagulation [28]. In the previous study, the alterations in the ROTEM parameters were related to concurrent hypothermia as well as to decreases in the platelet count and in Hct. There were differences in the blood collection times between studies: 3.5 h after the induction of anesthesia in the previous study and 24 h after surgery in our two studies. We hypothesise that the impact of anaesthesia and fluid therapy might decrease when blood samples are collected 24 h after surgery.

Hypercoagulability has also been identified in the post-operative period in some human studies, but unlike the results obtained in our dog population, the alterations are not transient and last for several weeks post-surgery. Evaluations of haemostasis in patients undergoing surgery for the removal of thoracic malignancies have identified that post-operative hypercoagulability persists for at least 1–2 weeks after surgery, whereas after the resection of intra-abdominal neoplasms, progressive hypercoagulability has been shown to persist for at least 1-month post-surgery [3,10]. In a study evaluating haemostasis after the resection of thoracoabdominal malignancies, coagulation returned to baseline levels at 6–12 months post-surgery [4].

Most methods for assessing the haemostasis (even TG) use plasma samples, while ROTEM uses whole blood, which allows the influence of both plasmatic and cellular components of blood to be considered during coagulation analysis. Therefore, ROTEM is theoretically the most suitable method for studying coagulation abnormalities in complex situations in which many factors are involved, such as in patients affected by neoplasia. However, modifications in the PLT number, fibrinogen, and Hct concentration influence the values of ROTEM parameters such as CFT, MCF, and the α angle, and to interpret the ROTEM profiles and their changes over time, we must take into account the haematological parameters.

From T0 to T1, there was a significant decrease in Hct and PLT and no difference in the fibrinogen concentration. The decrease of Hct induces a faster clot formation and an increase of clot stiffness [29,30]. The correlation between a decrease in Hct and a tendency towards hypercoagulability during ROTEM tracing depends on a change in the ratio between the corpuscolar and plasmatic parts of blood. In anaemia, the plasmatic component increases, causing a relative increase in the fibrinogen content [31]. This tendency towards hypercoagulability could be partly due to change in the Hct concentration, but the authors belive that surgery and the related systemic consequences are the main thrombophilic mechanisms. Indeed, no differences in Hct were observed between T1 and T2, but both time points showed significant decreases in Hct compared to at T0. Statistically significant differences were detected for several ROTEM parameters between T0 and T1 and for only one parameter (CT in the fib-TEM profile) between T0 and T2. This reasoning does not rule out the effect of a decrease in the Hct value on detected haemostatic changes but raises the suspicion that there are also other factors that could have been influenced. A recent study conducted on healthy dogs to evaluate the effects of anaesthesia on coagulation observed that 3.5 h after the induction of anaesthesia, few ROTEM parameters were significantly different from the pre-operative values [28]. In particular, only CT in the exTEM and fibTEM profiles was decreased, and MCF in the fibTEM profile was increased, but the changes were considered clinically irrelevant by the authors [BIBLIO]. These alterations were mainly attributed to a decrease in Hct (from 49% to 37%) and to a decrease in the platelet count [28]. In our study, the alterations that were found to have taken place involved several ROTEM parameters, indicating that other factors (inflammation, tissue trauma, and neoplasms) could have contributed to the alterations observed in the ROTEM tracings.

Another consideration is that the alterations detected in vitro with the ROTEM tool cannot reflect what actually happens in vivo. Indeed, the decrease in Hct may lead to an increase in bleeding time in vivo as a result of the rheological changes involving a decrease in platelet marginalization and consequent functional thrombocytopathy [32]. A study conducted in human beings showed an increase in the bleeding time of 60% following a 15% decrease in Hct, while this alteration did not occur when the PLT count decreased [32]. The authors therefore concluded that in haemodiluted patients with reduced Hct levels, decreased PLT counts, and reduced clotting protein levels, the administration of red blood cells to achieve Hct values above 34% was adequate to reduce both the bleeding time and nonsurgical blood loss, without the need to transfuse platelets or fresh frozen plasma [32]. In our population of dogs, the decrease in Hct between T0 and T1 was 8.6%, and only two dogs had an Hct <35% at T1. Thus, it can be speculated that in this case, the decrease in Hct may have had a limited impact on in vivo bleeding impairment.

A decrease in the PLT count causes less rapid clot formation and a decrease in clot stiffness, but that influence cannot be seen in the ROTEM results because at T1, the significant changes were an increase of MCF, the α angle, and G [29,30].

The decreases in the Hct and the PLT count between T0 and T1 can be explained by several concurrent mechanisms involved in primis blood loss during surgery, but unfortunately, data regarding surgical blood loss and total solids are not available. Another explanation might be the fluid therapy administered during the procedure, which may have diluted the concentration of erythrocytes and the PLT to a certain degree [33,34]. Finally, an anaesthetic-induced decrease in hematocrit up to 20% is demonstrated under halothane anaesthesia, a consequence of the erythrocyte sequestration in the spleen and in the splanchnic vasculature [35]. We cannot exclude that other types of anaesthetic drugs may also affect the Hct, but in the present study, the blood samples were collected 24 h after surgery, which may decrease the impact of anaesthesia.

From T1 to T2, a significant decrease in the fibrinogen concentration was identified. The decrease in the fibrinogen concentration led to a decrease in the MCF value in the ex-TEM and fib-TEM profiles [4,29]. This difference reflects the decrease in systemic inflammation and could partly explain the decrease in ROTEM parameters such as MCF and the α angle observed 2 weeks after surgery in the ex-TEM and fib-TEM profiles.

In human medicine, the application of TG in cancer patients have resulted in the detection of an association between thrombin increase and deep vein thrombosis events and as well as an increase in thrombin production in some patients [36,37,38]. In our study, the haemostasis assessments performed by means of thrombin generation showed no significant differences among the data obtained before surgery (T0), 24 h after surgery (T1), and 2 weeks (T2) after surgery.

The present study has some limitations. The number of animals included was adequate to show the differences between the times of the sample selected, but it would have been interesting to increase the number of dogs to create different groups. Differentiating dogs with cancer on the basis of haemostasis abnormalities will make it possible to observe and compare coagulation trends in these two groups and to better understand the clinical implications of the alterations detected. Increasing the number of dogs included could also allow the creation of subgroups of dogs to differentiate between types of malignancy or the types of surgery performed.

Due to the effect of Hct on ROTEM, we cannot exclude that haemostatic alterations detected in vitro accurately reflect what happens in vivo.

This study did not have a control group, but two recent studies have assessed the effect of elective surgery or anaesthesia on haemostasis in healthy dogs, observing few changes in the ROTEM parameters compared to several alterations observed in the present study [27,28].

Another limitation is not having performed additional coagulation assessments, such as those to determine individual plasmatic factors and anticoagulants. These additional evaluations could allow a better understanding of the haemostatic alterations that occur during surgery.

## 5. Conclusions

Surgery performed in dogs with resectable malignancies to remove cancer affects coagulation, causing a trend towards hypercoagulability. Two weeks after surgery, there was a reversal of the hypercoagulable tendency, but two dogs that were classified as hypercoagulable at T0 were still hypercoagulable at T2. The results of this study observed a transient influence of surgery on haemostasis and may help us to understand how long anticoagulant treatments should be administered in dogs affected by cancer and undergoing surgery.

Future research should focus on evaluating coagulation trends in hypercoagulable, hypocoagulable, and normocoagulable dogs with cancer using separated groups and prolonging post-operatory follow up to see long-term haemostatic alterations. Improving our knowledge of haemostasis in dogs with cancer will allow us to determine more appropriate guidelines for antithrombotic treatment and to avoid treats that are not useful and expensive.

## Figures and Tables

**Figure 1 animals-13-00164-f001:**
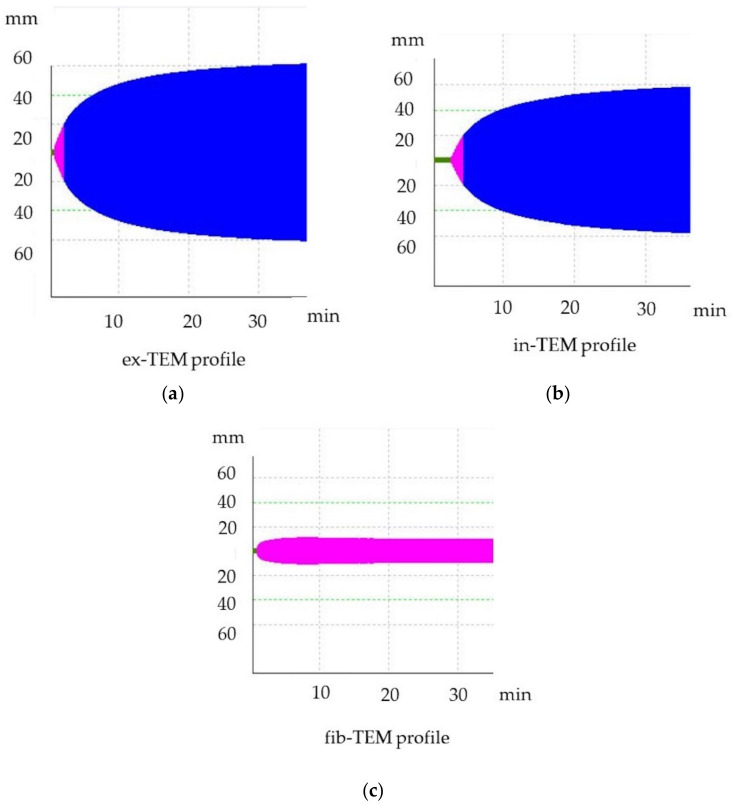
Thromboelastometric tracing of healthy dogs. (**a**) ex-TEM, extrinsic thromboelastometry pathway; (**b**) in-TEM, intrinsic thromboelastometry pathway; (**c**) fib-TEM, functional fibrinogen.

**Figure 2 animals-13-00164-f002:**
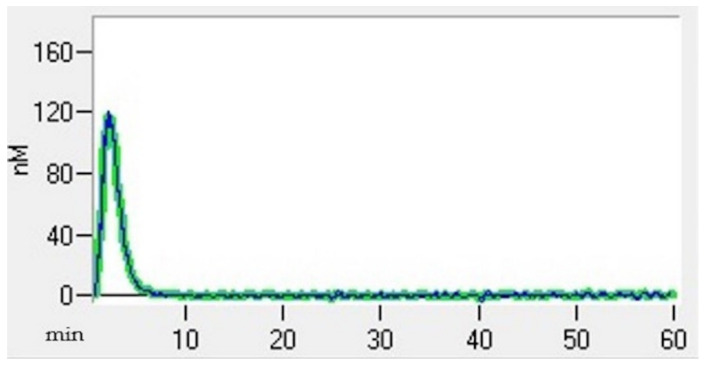
Thrombin generation curve of a healthy dog.

**Figure 3 animals-13-00164-f003:**
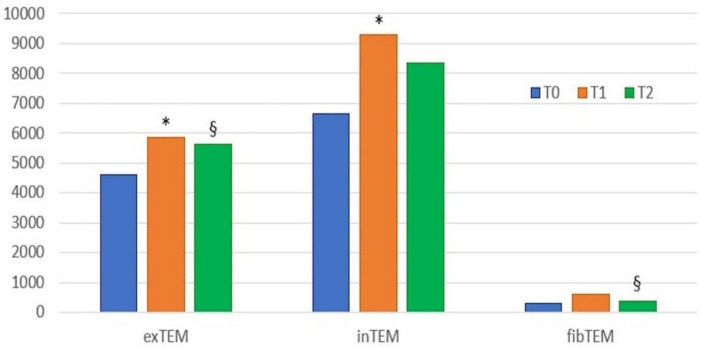
Histogram of the results (median values) of ROTEM parameter G at the three time points. T0, blood samples collected before surgery; T1, blood collected 24 h after surgery; T2, blood collected two weeks after surgery; * statistically significant differences between T0 and T1; ^§^ statistically significant differences between T1 and T2. A value of *p* < 0.05 indicates statistically significant differences. In-TEM, intrinsic thromboelastometry pathway; ex-TEM, extrinsic thromboelastometry pathway; fib-TEM, functional fibrinogen; CT, clotting time; CFT, clot formation time; MCF, maximum clot firmness; G = 5000 × MCF/(100 − MCF).

**Figure 4 animals-13-00164-f004:**
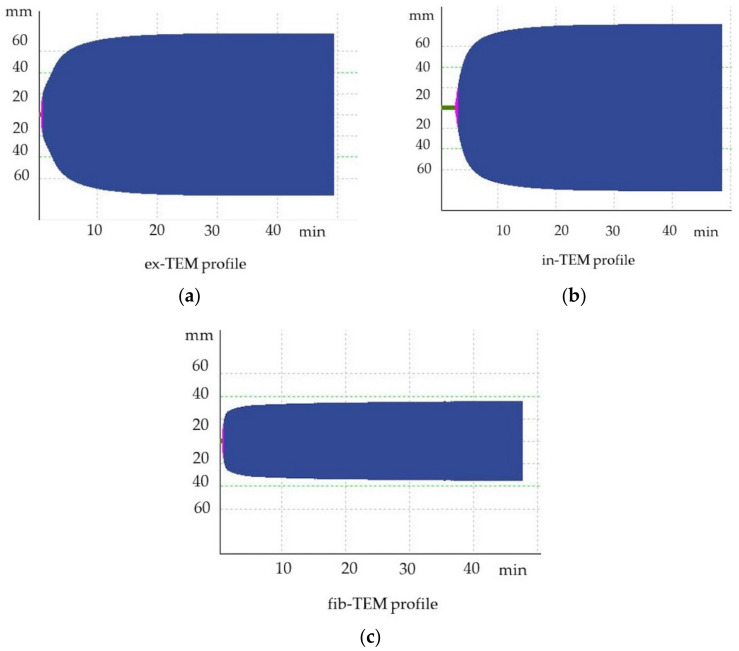
Hypercoagulable thromboelastometric tracing of a dog after bilateral maxillectomy for osteosarcoma (T1). (**a**) ex-TEM, extrinsic thromboelastometry pathway; (**b**) in-TEM, intrinsic thromboelastometry pathway; (**c**) fib-TEM, functional fibrinogen.

**Figure 5 animals-13-00164-f005:**
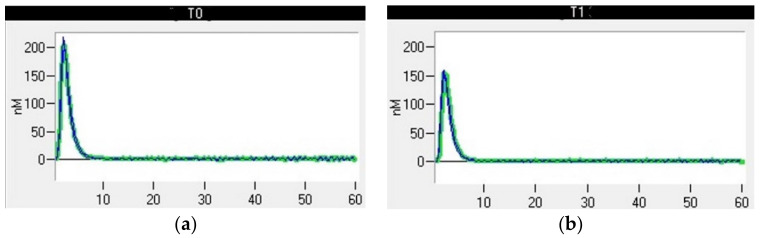
Thrombin generation curve of a dog after bilateral maxillectomy for osteosarcoma. (**a**) T0, thromboelastometric tracing performed with blood samples collected before surgery; (**b**) T1, thromboelastometric tracing with blood collected 24 h after surgery.

**Table 1 animals-13-00164-t001:** Thromboelastometric results assessed before surgery (T0), 24 h after surgery (T1), and two weeks after surgery (T2).

	T0 N = 11	T1 N = 11	T2 N = 10	*p* Value *	*p* Value ^#^	*p* Value ^§^	Institutional Reference Values
ROTEM							
**In-TEM**							
CT (s)	139 (112–182)	147 (108–172)	137 (121–242)	0.11	0.85	0.64	128–210 s
CFT (s)	97 (33–169)	68 (29–130)	81 (37–270)	0.032	0.36	0.14	58–200 s
MCF (mm)	57 (47–79)	65 * (56–81)	60.5 (48–75)	0.006	0.64	0.14	50–72 mm
α angle (°)	71 (58–84)	76 (67–84)	74.5 (46–82)	0.06	0.36	0.34	57–80°
G	6628 (4434–18,809)	9286 * (6364–21,315)	8336 (4615–15,000)	0.02	0.64	0.96	5417–12,119
**Ex-TEM**							
CT (s)	36 (25–78)	34 (24–56)	39 (26–59)	0.09	0.5	0.14	24–87 s
CFT (s)	182 (107–332)	151 (105–291)	150 ^§^ (75–304)	0.27	0.09	0.019	85–355 s
MCF (mm)	48 (37–80)	54 * (43–77)	53 ^§^ (40–65)	0.019	0.96	0.035	33–64 mm
α angle (°)	56 (43–87)	71 * (48–87)	61 ^§^ (43–83)	0.035	0.14	0.019	41–78°
G	4615 (2937–20,000)	5870 * (3771–16,739)	5638 ^§^ (3333–9287)	0.006	0.14	0.035	2253–5927
**Fib-TEM**							
CT (s)	41 (24–64)	32 * (26–42)	34.5 ^#^ (24–47)	0.033	0.0039	0.77	20–110s
MCF (mm)	5 (3–36)	11 (6–35)	7 ^§^ (4–17)	0.054	0.36	0.035	3–9 mm
G	289 (155–2813)	618 (319–2662)	376 ^§^ (208–1024)	0.055	0.36	0.035	113–509

* Statistically significant difference between T0 and T1; ^#^ Statistically significant difference between T0 and T2; ^§^ Statistically significant difference between T1 and T2. A value of *p* < 0.05 indicates statistically significant differences. Values are expressed as the median (minimum–maximum). T0, blood samples collected before surgery; T1, blood collected 24 h after surgery; T2, blood collected two weeks after surgery; in-TEM, intrinsic thromboelastometry pathway; ex-TEM, extrinsic thromboelastometry pathway; fib-TEM, functional fibrinogen; CT, clotting time; CFT, clot formation time; MCF, maximum clot firmness; G = 5000 × MCF/(100 − MCF).

**Table 2 animals-13-00164-t002:** Haemostasis assessment and laboratory parameters of interest assessed before surgery (T0), 24 h after surgery (T1), and two weeks after surgery (T2).

	T0 N = 11	T1 N = 11	T2 N = 10	*p* Value *	*p* Value ^#^	*p* Value ^§^	Institutional Reference Values
**Lagtime** (min)	1 (1–1.7)	1.2 (1–2.3)	1 (1–1.7)	0.99	0.5	0.38	
**ETP** (nM.min)	265.6 (218.3–439)	245 (153.9–334.6)	243 (174.7–475.8)	0.11	0.69	0.45	
**Peak** (nM)	98.5 (69.9–178.6)	96.5 (56.2–140.1)	92.6 (56.2–150.6)	0.75	0.69	0.45	
**ttPeak** (min)	2.75 (2.3–4)	2.7 (2.2–4.5)	2.7 (2.3–3.7)	0.99	0.5	0.38	
Standard coagulation							
**aPTT** (s)	13 (11–14)						12–16 s
**PT** (s)	8 (7.3–8.6)						8–10 s
**Fibrinogen** (μmol/L)	6.6 (3.9–19.8)	9.4 (6.6–17)	5.6 ^§^ (4.1–11)	0.065	1	0.019	4.4–13.2 (μmol/L)
Laboratory parameters							
**Hct** (%)	47.7 * (37.1–56.7)	41.3 (25.7–60.2)	44.3 ^#^ (34.8–57)	0.006	0.039	0.29	37.5–58.3%
**Platelet count** **(×10^9^cells/L)**	326 (210–510)	314 * (156–439)	422 ^#,§^ (229–590)	0.039	0.002	0.002	128–543 ×10^9^ cells/L (128–543 × 10^3^ cells/µL)
**PU/CU**	0.18 (0.11–1.93)						

* Statistically significant difference between T0 and T1; ^#^ Statistically significant difference between T0 and T2; ^§^ Statistically significant difference between T1 and T2. A value of *p* < 0.05 indicates statistically significant differences. Values are expressed as the median (minimum–maximum). T0, blood samples collected before surgery; T1, blood collected 24 h after surgery; T2, blood collected two weeks after surgery; ETP, endogen thrombin potential; ttPeak, time to peak; PT, prothrombin time; aPTT, activated partial thromboplastin time; Hct, haematocrit; PU/CU, urinary protein and urinary creatinine ratio.

## Data Availability

All data analysed during this study are included in this published article.

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
