# Peer review of "Evaluation of Haemostasis in Dogs Affected by Resectable Malignancy"

_animals, 2022, doi:10.3390/ani13010164_

Round 1
Reviewer 1 Report
this is an interesting report of the use of TEG for monitoring coagulation profiles in dogs with a variety of neoplasms. The manuscript could be improved by adding more graph/ figures in the results rather than the large table that you currently have. The major concern is that there are no "non neoplastic Healthy control animal" data provided. Need this to establish the effects of the surgical procedures and anesthesia if any on the Coag. results especially at T1 in dogs with no neoplasia. Fig 1 shows a normal dog TEG tracing It would improve the paper if you could include within this figure or another figure tracings for the affected dogs related to this at the different times based on the Table. with range bars. The ranges for each parameter are wide. In the introduction paragraph next to the last there is a missing reference (the sentence following ref 16-17 ). Fig 2 would also benefit from the addition of data from the neoplasia dogs as well as the single control dog shown
Author Response
All changes made in the text are in red color. The text was submitted to editing service (MDPI Language Editing Service) for extensive English revisions.
Reviewer 1
Comments and Suggestions for Authors
The manuscript could be improved by adding more graph/ figures in the results rather than the large table that you currently have.
Answer: Authors agree with the reviewer that a graph might help the reader to understand the results. A histogram of the results (median values) of ROTEM parameter G at the three time points, was added to the text.
The table is mandatory to report the results. We tried to break it into smaller tables, but it seemed to us that it made the text more difficult to read. If the reviewer has other suggestion to improve the presentation of results, authors will consider it carefully.
The major concern is that there are no "non neoplastic Healthy control animal" data provided. Need this to establish the effects of the surgical procedures and anesthesia if any on the Coag. results especially at T1 in dogs with no neoplasia
Answer: The present study doesn’t have a control group, but the authors have conducted a previous study on dogs undergoing orthopaedic surgery (adult or young animals, without underlying pathology) to observe the influence of surgery on hemostasis. Blood samples for ROTEM analysis were performed 1 hour before the surgery, 24 hours after the conclusion of the surgery, and 7 days after surgery. In the cited study, only few ROTEM parameters have changed after elective surgery, such as in another recent paper conducted on healthy dogs, that evaluate the effect of anaesthesia on coagulation. The few ROTEM alterations detected were related to concurrent hypothermia, decrease in platelet count and Hct, but unlike our two studies the blood samples were collected 3.5 hours after the induction of anesthesia. The authors hypothesize that the impact of anaesthesia and fluid-therapy might decrease when the blood samples were collected after 24 hours from surgery.
This previous explanation and relative references were added to the discussion. Please, see the text.
Fig 1 shows a normal dog TEG tracing It would improve the paper if you could include within this figure or another figure tracings for the affected dogs related to this at the different times based on the Table. with range bars.
Answer: The ROTEM tracings of the hypercoagulable dog undergoing surgery for bilateral maxillectomy for osteosarcoma (T1), was added in the text.
The ranges for each parameter are wide.
Answer: Authors agree with the reviewer that results are wide, but they are similar to other ROTEM ranges reported in literature and related to the viscoelastic methods.
Institutional reference values for ROTEM parameters (except G values) were previously established at our institution in 45 healthy dogs and published in the following study: Falco S. et al. In vitro evaluation of canine hemostasis following dilution with hydroxyethyl starch (130/0.4) via thromboelastometry. J Vet Emerg Crit Care. 2012;22(6):640-5. doi: 10.1111/j.1476-4431.2012.00816.x
In the introduction paragraph next to the last there is a missing reference (the sentence following ref 16-17 ).
Answer: Missing references were added to the text. Please, see the text.
Fig 2 would also benefit from the addition of data from the neoplasia dogs as well as the single control dog shown
Answer: The thrombin generation tracing of the hypercoagulable dog undergoing surgery for bilateral maxillectomy for osteosarcoma (T0 and T1), was added in the text.

Reviewer 2 Report
This is a small study involving 11 dogs undergoing surgery because of solid tumours. The dogs are of different breeds, diagnosed with different types of cancer and type of surgery varies extensively within the cohort. Laboratory values reflecting coagulation have been sampled at 3 time points, before, 24h after and 2 weeks after surgery and comparisons have been made inbetween time points, no control group has been involved. Despite the wide variety of diagnosis and the extremely small cohort, significant changes have been observed reflecting a transient hypercoagulability apparent at 24 hours after surgery that is diminished 2 weeks after surgery measured by the viscoelastic method ROTEM. The paper is clearly written, but multiple spelling-and grammatical errors are present (hightlighted with yellow), reducing the overall impression. The description of the ROTEM and TG methods were very nicely conducted and helpful when interpreting the results.
Minor comments
-The thrombin generation assay is very sensitive to whether platelets are present or not and whether the sample has been frozen or not. This information is i lacking in the "Methods" section
-Parameters such as HcT and plt count is highly infuenced by blood loss during surgery. Data on blood loss is needed to correctly interpret the results of the laboratory evaluation
My major comment is that I miss a discussion in how the coagulation system acts in viscoelastic methods/thrombin generation compared to the in vivo situation. ROTEM has the advantage of using whole blood, thus all cells present in whole blood also influence the result. This is to the contrary of TG where platelet poor or platelet rich plasma is used. Thus, ROTEM theoretically is the better method to use in a complicated setting such as cancer, affecting all blood cells as well as the plasma coagulation. The major disadvantage with ROTEM is the effect of Hct. As described in the paper a decreased heamatocrit will result in a faster clot formation and an increase of clot stiffness. This is to the contrary of the in vivo situation where decreased Hct induces less marginalisation of platelets to the vessel wall and thus, a functional thrombocytopenia. This fact makes the ROTEM results difficult to interpret in a situation where the Hct has varied significnatly over the study period. I would ask the authors to discuss these limitations and how they could be overcome

Author Response
Comments and Suggestions for Authors
This is a small study involving 11 dogs undergoing surgery because of solid tumours. The dogs are of different breeds, diagnosed with different types of cancer and type of surgery varies extensively within the cohort. Laboratory values reflecting coagulation have been sampled at 3 time points, before, 24h after and 2 weeks after surgery and comparisons have been made inbetween time points, no control group has been involved. Despite the wide variety of diagnosis and the extremely small cohort, significant changes have been observed reflecting a transient hypercoagulability apparent at 24 hours after surgery that is diminished 2 weeks after surgery measured by the viscoelastic method ROTEM. The paper is clearly written, but multiple spelling-and grammatical errors are present (hightlighted with yellow), reducing the overall impression. The description of the ROTEM and TG methods were very nicely conducted and helpful when interpreting the results.
Answer: Authors are very grateful to the reviewer for reviewing the manuscript. Authors hope to have clearly answered to all questions. All changes made in the text are in red color.
The paper is clearly written, but multiple spelling-and grammatical errors are present (hightlighted with yellow), reducing the overall impression.
Answer: Spelling and grammatical errors highlighted with yellow have been corrected and the text was submitted to editing service (MDPI Language Editing Service) for extensive English revisions.
Minor comments
-The thrombin generation assay is very sensitive to whether platelets are present or not and whether the sample has been frozen or not. This information is i lacking in the "Methods" section
Answer: To performed the thrombin generation authors have used citrated platelet poor plasma, stored at -80°C for less than 12 months. The information and relative reference were added to the “materials and methods”. Please, see the text.
-Parameters such as HcT and plt count is highly infuenced by blood loss during surgery. Data on blood loss is needed to correctly interpret the results of the laboratory evaluation
Answer: Data regarding surgical blood loss are not available. Despite this, we can hypothesize that the decrease in hematocrit is due to blood loss, partly to hemodilution and cannot be excluded an impact of anesthesia. Authors have added this concept in the discussion and in the limitation of the study.
My major comment is that I miss a discussion in how the coagulation system acts in viscoelastic methods/thrombin generation compared to the in vivo situation. ROTEM has the advantage of using whole blood, thus all cells present in whole blood also influence the result. This is to the contrary of TG where platelet poor or platelet rich plasma is used. Thus, ROTEM theoretically is the better method to use in a complicated setting such as cancer, affecting all blood cells as well as the plasma coagulation. The major disadvantage with ROTEM is the effect of Hct. As described in the paper a decreased heamatocrit will result in a faster clot formation and an increase of clot stiffness. This is to the contrary of the in vivo situation where decreased Hct induces less marginalisation of platelets to the vessel wall and thus, a functional thrombocytopenia. This fact makes the ROTEM results difficult to interpret in a situation where the Hct has varied significnatly over the study period. I would ask the authors to discuss these limitations and how they could be overcome
Answer: Authors are very grateful for this suggestion, as it has improved the quality of the discussion. The text was modified and implemented in the discussion and limits sections, to account the reviewer suggestions. Please, see the text.
